

# Purpureocillium lilacinum and Metarhizium marquandii as plant growth-promoting fungi

Noemi Carla Baron[1], Andressa de Souza Pollo[2] and Everlon Cid Rigobelo[1]

[1] Agricultural and Livestock Microbiology Graduation Program, São Paulo State University (UNESP), School of Agricultural and Veterinarian Sciences, Jaboticabal, São Paulo, Brazil
[2] Department of Preventive Veterinary Medicine and Animal Reproduction, São Paulo State University (UNESP), School of Agricultural and Veterinarian Sciences, Jaboticabal, São Paulo, Brazil

Corresponding author
Everlon Cid Rigobelo,
everlon.cid@unesp.br

## ABSTRACT

**Background:** Especially on commodities crops like soybean, maize, cotton, coffee and others, high yields are reached mainly by the intensive use of pesticides and fertilizers. The biological management of crops is a relatively recent concept, and its application has increased expectations about a more sustainable agriculture. The use of fungi as plant bioinoculants has proven to be a useful alternative in this process, and research is deepening on genera and species with some already known potential. In this context, the present study focused on the analysis of the plant growth promotion potential of *Purpureocillium lilacinum*, *Purpureocillium lavendulum* and *Metarhizium marquandii* aiming its use as bioinoculants in maize, bean and soybean.

**Methods:** *Purpureocillium* spp. and *M. marquandii* strains were isolated from soil samples. They were screened for their ability to solubilize phosphorus (P) and produce indoleacetic acid (IAA) and the most promising strains were tested at greenhouse in maize, bean and soybean plants. Growth promotion parameters including plant height, dry mass and contents of P and nitrogen (N) in the plants and in the rhizospheric soil were assessed.

**Results:** Thirty strains were recovered and characterized as *Purpureocillium lilacinum* (25), *Purpureocillium lavendulum* (4) and *Metarhizium marquandii* (1). From the trial for P solubilization and IAA production, seven strains were selected and inoculated in maize, bean and soybean plants. These strains were able to modify in a different way the evaluated parameters involving plant growth in each crop, and some strains distinctly increased the availability of P and N, for the last, an uncommon occurrence involving these fungi. Moreover, the expected changes identified at the in vitro analysis were not necessarily found *in planta*. In addition, this study is the first to evaluate the effect of the isolated inoculation of these fungi on the growth promotion of maize, bean and soybean plants.

## INTRODUCTION

As it is in many other countries, agriculture is the main activity of the Brazilian economy. Brazil became the largest soybean producer worldwide in 2018 (*Compania Nacional de Abastecimento (CONAB), 2019*) and is the third largest producer of maize (*Compania Nacional de Abastecimento (CONAB), 2016*) and beans (*Compania Nacional de Abastecimento (CONAB), 2017*). Soybean and maize are the main grains exported, while beans are consumed within Brazil as a food source. Currently, due to awareness about the environmental and health risks from the excessive use of pesticides and fertilizers, agriculture is undergoing a transformation process, and alternatives for more sustainable management are being developed. In this context, microorganisms have gained increasing prominence due to the success of their current application and the promising results from innovative research in this field.

Associations with microorganisms favor the survival of plants in the environment for several reasons. Plant mutualistic fungi are currently recognized as a new and important source of bioactive compounds. They produce a significant number of secondary metabolites, including phytohormones (auxins, gibberellins, cytokinins, and abscisic acid) and antifungal and antibacterial compounds (*Rai et al., 2014*). Studies have shown that plants treated with a variety of symbiotic fungi are often healthier than untreated plants (*Strobel & Daisy, 2003*; *Hyde & Soytong, 2008*; *Khan et al., 2008*; *Colla et al., 2015*).

Such plant-fungus associations are established mainly by two groups of fungi, mycorrhizal and endophytic fungi (*Bonfante & Genre, 2010*). Endophytic fungi are those capable of living endosymbiotically with plants without causing disease symptoms (*Behie & Bidochka, 2014a*). They can act as plant growth promoters, increase the germination rate, improve seedling establishment, and increase plant resistance to biotic and abiotic stresses by producing antimicrobial compounds, phytohormones and other bioactive compounds. In addition, endophytic fungi are responsible for the acquisition of soil nutrients, including macronutrients such as phosphorus, nitrogen, potassium and magnesium, and micronutrients such as zinc, iron, and copper (*Behie & Bidochka, 2014a*; *Rai et al., 2014*; *Khan et al., 2015*).

Among the several fungal species capable of endophytically colonizing plants, this study focuses on the application of *Purpureocillium* (formerly *Paecilomyces*) strains and *Metarhizium marquandii* (formerly *Paecilomyces marquandii*). Studies have shown that fungi of the genus *Purpureocillium* and other closely related fungi have attributes that promote plant growth; however, only a few studies have presented these results *in planta*.

Taxonomically, the genus *Paecilomyces* was introduced by *Bainier (1907)*. It is a polyphyletic genus, with representatives in distinct families and even orders. In recent years, it has undergone extensive revision, which has resulted in several taxonomic changes. One of these was proposed by *Luangsa-ard et al. (2011)*, who described an in-depth morphological and phylogenetic approach to *Paecilomyces* species. They proposed the genus *Purpureocillium* to accommodate the species *Paecilomyces lilacinus*, modifying it to *Purpureocillium lilacinum*, which is commonly found in soil and is one of the most studied nematophagous fungi (*Baron, Rigobelo & Zied, 2019*). Some authors

have already studied a large number of biological nematicides based on *P. lilacinum* strains (*Atkins et al., 2005*; *Dong & Zhang, 2006*; *Baron, Rigobelo & Zied, 2019*). In many cases, these biological nematicides can replace chemical nematicides, which are nonspecific and nonselective, do not affect the development of worm eggs, are expensive, and are toxic to harmless species of invertebrates and vertebrates, including humans (*Li et al., 2015*; *Degenkolb & Vilcinskas, 2016*).

In addition to the potential of *P. lilacinum* as a biocontrol agent against nematodes, there is scarce information about this species, others from this genus, or any other formerly recognized as *Paecilomyces* in terms of plant growth promotion. Some studies have already described *Paecilomyces/Pupureocillium* species in plants as endophytes (*Bills & Polishook, 1991*; *Cao, You & Zhou, 2002*; *Tian et al., 2004*), indicating their ability to colonize plants. Thus, these fungi may be able to establish different interactions with plants to benefit themselves and their host. Therefore, this study aimed to isolate strains formerly recognized as *Paecilomyces*, mainly *Purpureocillium* strains, from soil; perform molecular characterization and in vitro tests to evaluate their abilities to solubilize P and produce IAA, which are characteristics related to growth promotion; and then select the best strains to be tested on maize (*Zea mays*), bean (*Phaseolus vulgaris*) and soybean (*Glycine max*) plants to assess their potential as biofertilizers.

## MATERIALS AND METHODS

### Fungal isolation

Fungal isolation was carried out from soil samples collected in an area of approximately 8.400 m$^2$ (~150 × 56 m) occupied by guava trees (*Psidium guajava*, Paluma variety, 173 individuals) and located in Taquaritinga town, São Paulo state, Brazil (GNSS: 21°22′20.9″S; 48°35′24.6″W Gr.). The guava trees in this area were highly infested with nematodes, which led to its use as a source of soil samples for *Paecilomyces/Purpureocillium* isolation. The area was divided into four similar quadrants. Five soil samples (up to 15 cm depth) were collected from each quadrant and mixed to obtain one composite sample for each quadrant.

In the laboratory, 10 g of each composite soil sample was taken and transferred to Erlenmeyer flasks containing 100 mL of sodium pyrophosphate solution (1 g/L). The samples were homogenized on a rotary shaker for 60 min at 90 rpm and then used for serial dilutions. One hundred microliter aliquots were plated on three different culture media: Martin's medium (*Martin, 1950*), DOC2 medium (*Shimazu & Sato, 1996*) and potato dextrose agar (PDA) supplemented with 1.5% sodium chloride (NaCl) (adapted from *Mitchell, Kannwischer-Mitchell & Dickson (1987)*). Martin's medium is a general medium for soil fungi, whereas PDA with NaCl and DOC2 are described as selective for *Paecilomyces/Purpureocillium* isolation. The plates were incubated at 25 °C for up to 21 days and checked daily for the emergence of new colonies.

The isolated cultures were purified, and morphological confirmation to the genus level was performed by growth analysis on culture media and microscopic observations of slides prepared with lactophenol blue dye. The cultures were preserved in slants containing PDA at 4 °C and by ultrafreezing at −80 °C with glycerol at 15% concentration.

## Molecular and phylogenetic analysis

For DNA extraction, the fungi were grown in small flasks containing 40 mL of potato dextrose broth (PDB) for 3–4 days at 26–28 °C. After incubation, the mycelium was recovered, washed with ultrapure water and oven dried at 50 °C for at least 12 h. The dried mycelium was macerated with liquid nitrogen and used for DNA extraction according to *Kuramae-Izioka (1997)*. Genomic DNA samples were purified with Wizard SV Gel and PCR Clean-up System (Promega, São Paulo) according to the manufacturer's specifications.

DNA amplification was performed for the nuclear ribosomal DNA ITS1-5.8S-ITS2 (ITS barcode) region with the primers ITS1 (5′ TCCGTAGGTGAACCTGCGG 3′) and ITS4 (5′ TCCTCCGCTTATTGATATGC 3′) (*White et al., 1990*) and for a fragment of the coding region of the β-tubulin protein (β*TUB2*) with the primers Bt2a (5′ GGTAACCAAA TCGGTGCTGCTTTC 3′) and Bt2b (5′ ACCCTCAGTGTAGTGACCCTTGGC 3′) (*Glass & Donaldson, 1995*).

The reactions were performed with 1X buffer (50 mM KCl, 200 mM Tris-HCl, pH 8.4); 2 mM MgCl$_2$; 0.2 mM dNTPs; 0.5 U Platinum Taq DNA polymerase (Invitrogen, Carlsbad, CA, USA); 2.5 pmol of each primer; 0.001 mg BSA; 0.1 μl DMSO; 60 ng of genomic DNA and sterile pure water to 20 μl. The amplification program was as follows: 95 °C for 4 min; followed by 35 cycles at 94 °C for 50 s, primer annealing temperature (55 °C for ITS1 and ITS4 and 58 °C for Bt2a and Bt2b) for 30 s, and 72 °C for 40 s; and a final extension at 72 °C for 10 min.

The PCR products were purified using the ExoSAP-IT PCR product cleanup kit (Applied Biosystems, Foster City, CA, USA) following the manufacturer's instructions. The sequencing reaction was performed with a BigDye Terminator kit (Applied Biosystems, Foster City, CA, USA) according to the manufacturer's instructions. Sequencing was performed by capillary electrophoresis on an ABI3130 sequencer.

The electropherograms were submitted to the PHRED/PHRAP/CONSED program package (*Green, 1996*; *Ewing et al., 1998*; *Gordon, Abajian & Green, 1998*). Only nucleotides with a Phred quality equal to 20 or higher were considered. The edited sequences were compared to those deposited in the National Center for Biotechnology Information (NCBI) GenBank database using the BLAST tool (*Altschul et al., 1990*) and in the CBS database (Centraalbureau voor Schimmelcultures–Westerdijk, Netherlands).

For phylogenetic analysis, the sequences of the ITS and β*TUB2* regions of the 30 strains and other sequences collected were aligned separately using the MUSCLE tool (*Edgar, 2004*). The best evolutionary model was selected according to the Akaike information criterion (AIC) using the MODELTEST version 3.7 tool (*Posada & Crandall, 1998*; *Posada & Buckley, 2004*). The sequences of the two regions were processed separately and subsequently concatenated for Bayesian analysis using the Markov Chain-Monte Carlo algorithm (MCMC) with the software MRBAYES version 3.2.3 (*Ronquist & Huelsenbeck, 2003*). The evolutionary model was used to analyze each region separately, and the concatenated sequences of the two genes corresponded to substitution type 6 and gamma

distribution. The phylogram obtained was graphically edited with DENDROSCOPE version 3 software (*Huson & Scornavacca, 2012*).

## Phosphorus solubilization and indoleacetic acid production assays

To verify the P solubilization ability of the fungi, Erlenmeyer flasks (125 mL) containing 50 mL P solubilization broth from *Nahas, Fornasieri & Assis (1994)* were prepared using fluorapatite (5 g/L) as the sole P source. To detect the IAA production ability, the 30 strains were grown in 125 mL Erlenmeyer flasks containing 50 mL of dextrose yeast glucose sucrose medium (DYGS) containing the amino acid tryptophan (*Milani et al., 2019*). In both tests, the flasks were inoculated with three disks of mycelium (8 mm diameter) cultured for 7 days on PDA and then incubated for 7 days on a rotary shaker at 25 °C ± 1 °C and 100 rpm.

   After incubation, the samples were vacuum-filtered using preweighed filter paper to separate the mycelial biomass. The amount of soluble P in the filtrate was determined by the *Ames (1966)*. For IAA detection, Salkowski reagent was added to the filtrate for the colorimetric reaction, and spectrophotometric readings were performed at 530 nm. Absorbance values were used to quantify soluble P and IAA production by comparison with standard curves with defined P and IAA concentrations. The filter papers with the biomass were oven-dried at 40 °C for 48 h and weighed again to obtain the dry biomass values. The data are presented as micrograms of P solubilized per gram of dried mycelium (µg P/g) and micrograms of IAA produced per gram of dried mycelium (µg IAA/g).

## Mass production of selected strains

The isolates selected in the screening tests (P solubilization and IAA production) were mass-grown on parboiled rice. The production process followed the *Alves & Pereira (1998)* methodology. The rice was inoculated with suspensions containing $10^6$ conidia/mL obtained by scraping cultured plates containing the selected strains grown on PDA for 7 days with 0.1% Tween 80 solution and standardized with a hematocytometer. The inoculation was performed by injecting approximately 5 mL of the suspensions into the rice with a needle and syringe. Rice colonization occurred at 25 °C for 15 days, and then the rice was transferred to plastic trays for drying for 3–5 days. The dried rice was packed in plastic bags and stored in a freezer at −20 °C.

## Greenhouse experiments

Selected strains were tested in maize (hybrid 2B587PW; Dow Agro Sciences, Indianapolis, IN, USA), bean (BRS FC402: Embrapa, Brasilia, Brazil) and soybean (Intacta RR2 Pro Monsanto, St. Louis, MO, USA) plants under greenhouse conditions.

   Five-liter pots were filled with unsterilized sifted ravine soil. A preliminary soil analysis was carried out, and nitrogen (N), phosphorus (P) and potassium (K) correction fertilization was prepared for planting following the guidelines of *Raij et al. (1997)* for each crop. Urea, simple superphosphate and potassium chloride were used as sources of N, P and K, respectively. Each pot received 5–6 seeds. After germination, thinning was

performed to maintain three plants per pot. A volume of approximately 250–300 mL of water was supplied once a day to each pot.

Inoculation was carried out with suspensions of each strain. The suspensions were prepared from the lavage of the colonized rice with 0.1% Tween 80 solution, and the concentrations were standardized with a hematocytometer to $10^8$ conidia/mL. Each pot received 20 mL of suspension. Two inoculations were performed in the soil: the first at sowing and the second 15 days after sowing. The viability of the conidia was evaluated at both inoculations by counting the number of conidia with germ tubes (%) after 17 h of incubation on PDA at 26 °C.

After 30 days, the pots were disassembled, and the following parameters were evaluated: (1) plant height: measurements were taken from the apex to the base for soybean and bean plants and from the apex of the third leaf to the base for maize plants; (2) dry mass: the plants were split into shoots and roots, dried in a forced-ventilation oven for 72–96 h at 65 °C and then weighed in a semianalytical balance; (3) P and N in the rhizospheric soil: a sample of rhizospheric soil was collected from each pot and used to evaluate the P content according to *Raij et al. (2001)*. For the soybean plants only, the N content in the soil was determined according to *Malavolta, Vitti & Oliveira (1997)*; (4) P and N in shoot and roots: dried shoot and root samples were ground and used for the determination of P and N contents. The P content determination was performed by nitric-perchloric digestion followed by spectrophotometric analysis (*Malavolta, Vitti & Oliveira, 1997*), and the N content was determined by sulfur digestion followed by titration (*Malavolta, Vitti & Oliveira, 1997*).

## Statistical analysis

The in vitro tests of P solubilization and IAA production were performed in a completely randomized design for the 30 strains, and each strain was considered a treatment with three replicates. The *F* test was carried out for variance analysis, followed by Tukey's test at 5% probability for comparison of the means.

The greenhouse experiment was carried out in a completely randomized design with four replicates (1 replicate = 1 pot), with each pot containing 3 plants. Data were submitted to variance analysis followed by Duncan's test at 5% probability for comparison of the means.

Statistical analysis was performed using AGROESTAT software (*Barbosa & Maldonado, 2015*).

# RESULTS

## Fungal isolation and molecular characterization

A total of 30 fungal strains morphologically recognized as *Paecilomyces/Purpureocillium* were obtained from the isolation. Of these 30 strains, two were obtained from Martin's medium, five from PDA supplemented with NaCl, and the remaining 23 from the DOC2 medium. In this study, the strains are referred to with the prefix LSM followed by the numerical code associated with the isolate (e.g., LSM 6).

The phylogenetic trees were developed with the ITS, β*TUB2* and concatenated sequences of the 30 strains obtained from this study and sequences from published strains and strains deposited in culture collections, mainly those from type strains of each species and species belonging to closely related families. They are presented in Figs. 1–3, respectively, and Table 1 lists all sequences used in the analysis with their respective GenBank accession numbers.

## In vitro screening tests and selection of strains for *in planta* tests

The results for fluorapatite solubilization by the 30 strains are shown in Fig. 4. The strains LSM 24, 14 and 65 of *P. lilacinum* were the three most effective P solubilizers from fluorapatite, with LSM 24 solubilizing the highest amount of P.

For IAA production, impressive results were obtained and are presented in Fig. 5. The strain LSM 65 of *P. lilacinum* produced more IAA than the other strains by a significant margin, followed by LSM 62 (*P. lilacinum*), LSM 68 (*M. marquandii*), LSM 183 and LSM 179 (*P. lilacinum*).

LSM 24, 14 and 65 were the best P solubilizers. Despite the primary statistical relevance of LSM 24, the two remaining strains were also selected because of their statistical relevance compared with the lowest P solubilizer strain in this test, LSM 73 (Fig. 4).

Regarding IAA production, strains LSM 65 and 62 were the best producers. However, using the same criterion applied in the P solubilization test, strains LSM 68, 179 and 183 were selected due to their statistical relevance compared with the lowest IAA producer, LSM 187.

Therefore, based on the screening tests, LSM 14, 24, 62, 65, 179 and 183 of *P. lilacinum* and LSM 68 of *M. marquandii* were selected for the *in planta* tests.

## Greenhouse tests with the selected strains

Figure 6 summarizes the significant results obtained from the statistical analysis of the evaluated parameters for maize, bean and soybean plants. All statistical data related to the three crops tested can be found in Tables S1–S3.

In maize plants, the *P. lilacinum* strain LSM 65 provided a significant increase in plant height compared with that in the control (Fig. 6A). The *P. lilacinum* strain LSM 24 significantly increased P levels in the soil compared with those of the control (66.7%) and the other strains (Fig. 6B).

The results observed in maize plants match the in vitro potential described for the tested fungi. LSM 65 was selected as the best IAA producer (Fig. 5) and was also a good P solubilizer, surpassed only by LSM 24 and 14 (Fig. 4). Although it did not stand out among the evaluated parameters involving P quantification, inoculation with this strain had a significant effect on the vegetative growth of the plant (shoot height), which was certainly related to IAA production. The best in vitro P solubilizer, LSM 24 (Fig. 4), showed high P solubilization under greenhouse conditions, which could be detected in the soil (Fig. 6B).

No significant differences were detected in N content in the shoot; however, in the roots, all the inoculated fungi were able to significantly increase the levels of N, except for

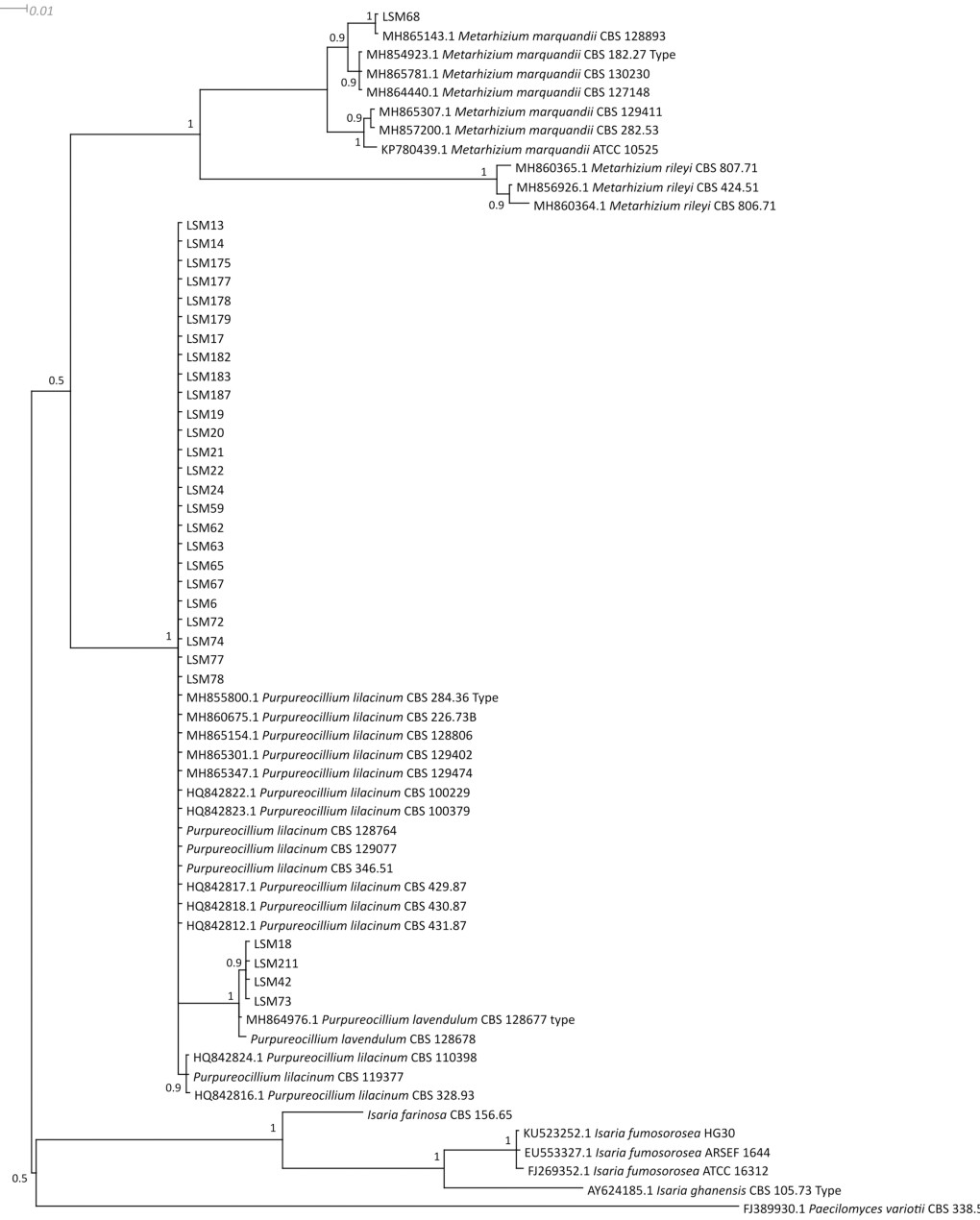

**Figure 1 Bayesian phylogenetic analysis of the ITS region of ribosomal DNA.** The sequences of this study are indicated by the LSM code. For the other sequences, the codes preceding the specific name refer to the GenBank accession number (NCBI). The sequences of the *P. lilacinum* strains CBS 128764, CBS 129077, CBS 346.51 and CBS 119377; *P. lavendulum* CBS 128678; and *I. farinosa* CBS 156.65 were collected from the CBS database. Type = Type strain.

LSM 14 (Fig. 6C). Notably, LSM 65 and 24 increased N levels in the roots by 39.2% and 38.0%, respectively, compared with that of the control.

For bean plants, unlike in maize, no significant differences in plant height, dry mass, N contents or P levels in the soil and roots were found in the treatments that received fungal inoculation (Table S2); however, significantly higher P levels were found in the
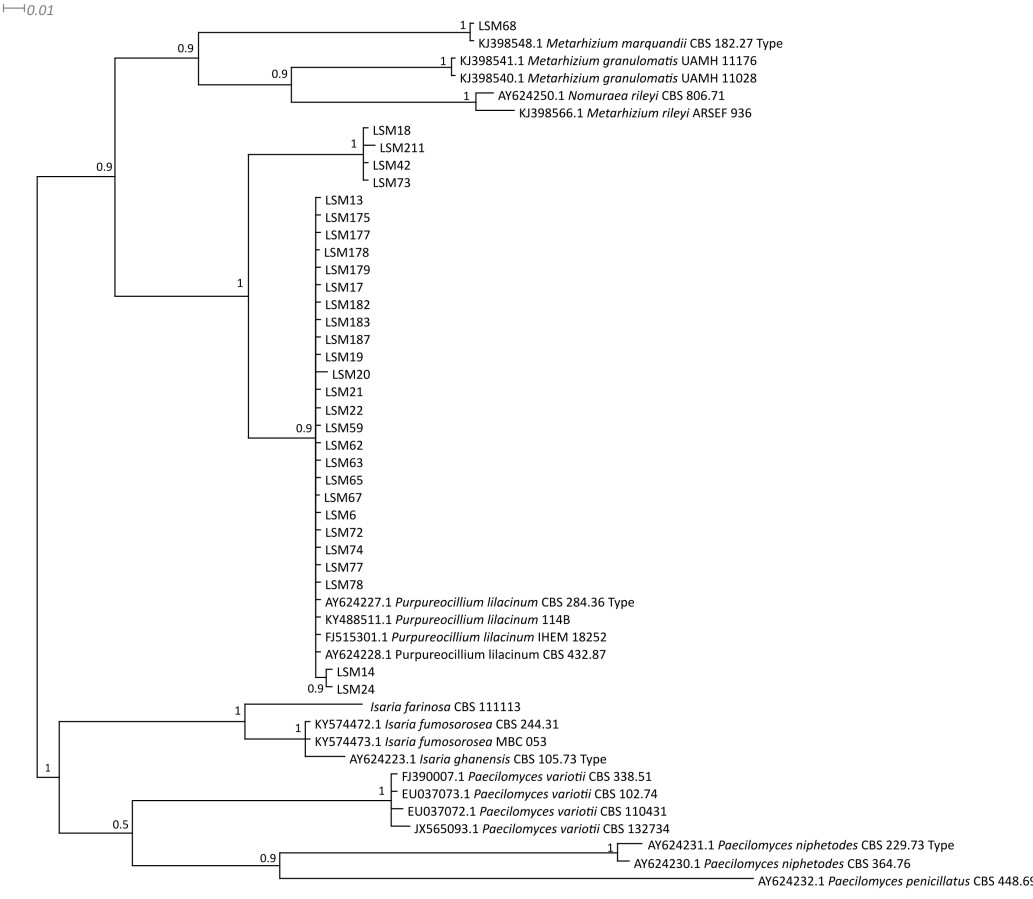

**Figure 2 Bayesian phylogenetic analysis of the β-tubulin coding region.** The sequences of this study are indicated by the LSM code. For the other sequences, the codes preceding the specific name refer to the GenBank accession number (NCBI). The sequence for the strain CBS 111113 was collected from the CBS database. Type = Type strain.

shoots of bean plants inoculated with LSM 68 (*M. marquandii*), LSM 179 (*P. lilacinum*) and LSM 183 (*P. lilacinum*), presenting 79.2%, 75.5% and 94.3% more P than in the control, respectively (Fig. 6D).

These data indicate the strong P solubilization ability of these strains, suggesting that the P was solubilized from the soil and that by direct absorption or by transportation through the hyphae, in the case of an endophytic interaction, the P reached the plants and was allocated to the shoot.

In soybean plants, LSM 179 promoted a significant increase in plant height compared to that of the control (Fig. 6E). For the variables related to dry mass, plants inoculated with LSM 24 had significantly higher root and total dry mass than those of the control (Figs. 6F and 6G).

In addition to increasing plant height, LSM 179 promoted a significant increase in N levels in the soil (Fig. 6H), more than doubling the amount of N found in the control treatment. However, for the shoot and root, none of the strains led to a significant increase

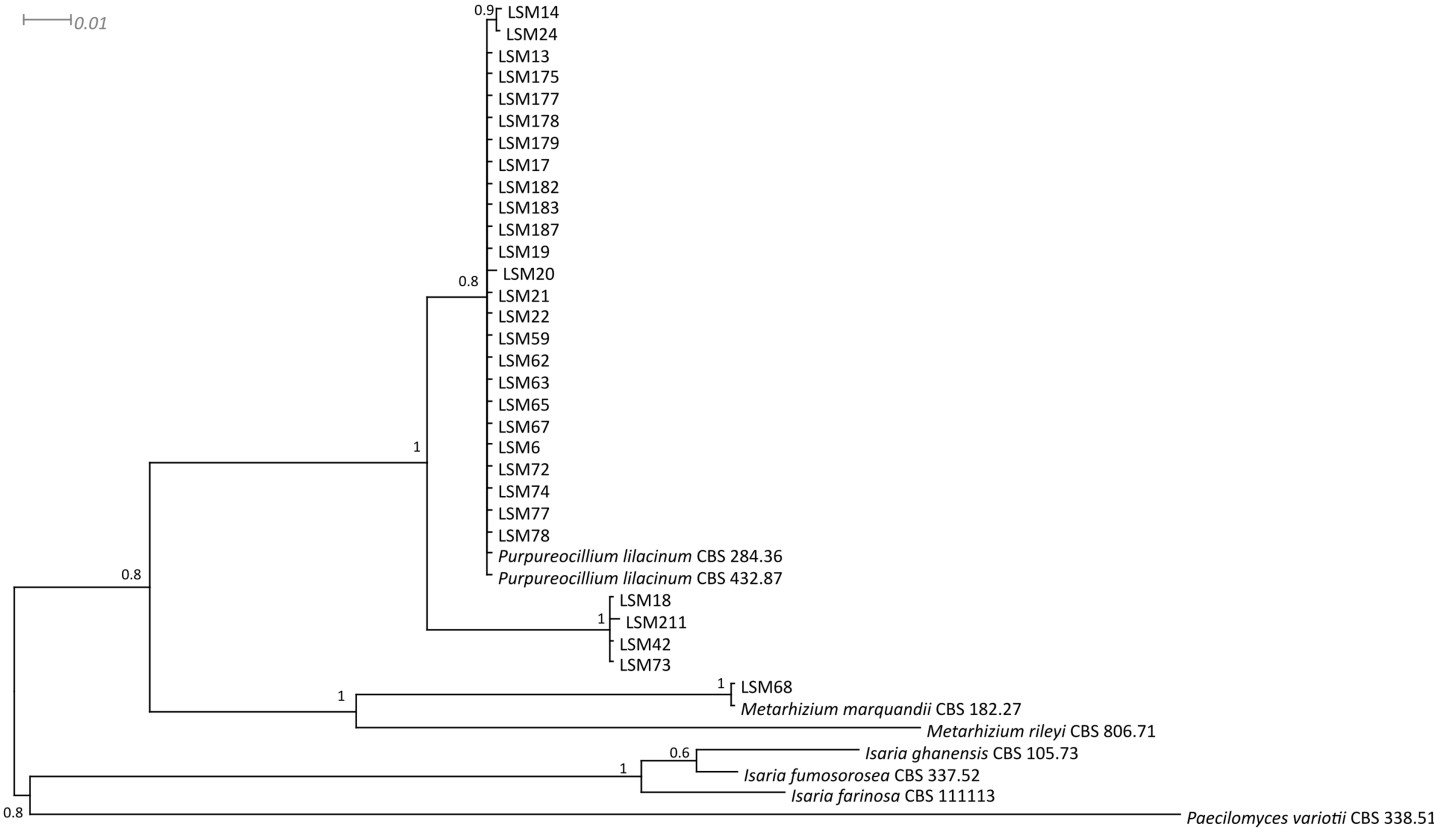

**Figure 3 Bayesian phylogenetic analysis of concatenated ITS and β-tubulin regions sequences.** The sequences in this study are indicated by the LSM code. The remaining sequences were collected from the CBS database.

in N levels. The evaluation of the P content in the soil and in the plants revealed that none of the strains promoted a significant increase in P content.

Despite such positive results, some negative interactions were observed and should be mentioned. For maize plants, LSM 183 was responsible for significant decreases in plant height and shoot, root and total dry mass compared with the control. Additionally, LSM 179 led to a reduction in the height of bean plants.

## DISCUSSION

### Isolation and molecular characterization

The use of selective media is common when the cultivation of a specific group is desired. In this study, two of the three media tested for fungal isolation were selective. Martin's medium (*Martin, 1950*) is generalist medium for soil fungi and was not demonstrated to be efficient for the recovery of the genus of interest. There are only a few media recommended in the literature for *Purpureocillium* (formerly *Paecilomyces*) isolation, and most of them include the addition of sodium chloride to PDA due to the characteristic osmotolerance of the genus. In this study, the recommendation of *Mitchell, Kannwischer-Mitchell & Dickson (1987)* to use PDA with 1.5% sodium chloride was adopted. However, the authors also suggested the use of PCNB (pentachloronitrobenzene)
**Table 1** List of all strains used in the phylogenies with their respective GenBank accession numbers for ITS and β-tubulin sequences.

| Cult. Collec. ID | Identification | GenBank accession number | |
| --- | --- | --- | --- |
| | | ITS | β-tubulin |
| LSM 6 | *Purpureocillium lilacinum* | MK506343 | MK550664 |
| LSM 13 | *Purpureocillium lilacinum* | MK506319 | MK550665 |
| LSM 14 | *Purpureocillium lilacinum* | MK506320 | MK550666 |
| LSM 17 | *Purpureocillium lilacinum* | MK506325 | MK550667 |
| LSM 18 | *Purpureocillium lavendulum* | MK506329 | MK550668 |
| LSM 19 | *Purpureocillium lilacinum* | MK506330 | MK550669 |
| LSM 20 | *Purpureocillium lilacinum* | MK506331 | MK550670 |
| LSM 21 | *Purpureocillium lilacinum* | MK506332 | MK550671 |
| LSM 22 | *Purpureocillium lilacinum* | MK506334 | MK550672 |
| LSM 24 | *Purpureocillium lilacinum* | MK506335 | MK550673 |
| LSM 42 | *Purpureocillium lavendulum* | MK506336 | MK550674 |
| LSM 59 | *Purpureocillium lilacinum* | MK506337 | MK550675 |
| LSM 62 | *Purpureocillium lilacinum* | MK506338 | MK550676 |
| LSM 63 | *Purpureocillium lilacinum* | MK506339 | MK550677 |
| LSM 65 | *Purpureocillium lilacinum* | MK506340 | MK550678 |
| LSM 67 | *Purpureocillium lilacinum* | MK506341 | MK550679 |
| LSM 68 | *Metarhizium marquandii* | MK506342 | MK550680 |
| LSM 72 | *Purpureocillium lilacinum* | MK506344 | MK550681 |
| LSM 73 | *Purpureocillium lavendulum* | MK506345 | MK550682 |
| LSM 74 | *Purpureocillium lilacinum* | MK506346 | MK550683 |
| LSM 77 | *Purpureocillium lilacinum* | MK506347 | MK550684 |
| LSM 78 | *Purpureocillium lilacinum* | MK506348 | MK550685 |
| LSM 175 | *Purpureocillium lilacinum* | MK506321 | MK550686 |
| LSM 177 | *Purpureocillium lilacinum* | MK506322 | MK550687 |
| LSM 178 | *Purpureocillium lilacinum* | MK506323 | MK550688 |
| LSM 179 | *Purpureocillium lilacinum* | MK506324 | MK550689 |
| LSM 182 | *Purpureocillium lilacinum* | MK506326 | MK550690 |
| LSM 183 | *Purpureocillium lilacinum* | MK506327 | MK550691 |
| LSM 187 | *Purpureocillium lilacinum* | MK506328 | MK550692 |
| LSM 211 | *Purpureocillium lavendulum* | MK506333 | MK550693 |
| CBS 284.36[T] | *Purpureocillium lilacinum* | MH855800 | AY624227 |
| CBS 226.73B | *Purpureocillium lilacinum* | MH860675 | - |
| CBS 128806 | *Purpureocillium lilacinum* | MH865154 | - |
| CBS 129402 | *Purpureocillium lilacinum* | MH865301 | - |
| CBS 129474 | *Purpureocillium lilacinum* | MH865347 | - |
| CBS 100229 | *Purpureocillium lilacinum* | HQ842822 | - |
| CBS 100379 | *Purpureocillium lilacinum* | HQ842823 | - |
| CBS 429.87 | *Purpureocillium lilacinum* | HQ842817 | - |
| CBS 430.87 | *Purpureocillium lilacinum* | HQ842818 | - |
| CBS 431.87 | *Purpureocillium lilacinum* | HQ842812 | - |
| CBS 110398 | *Purpureocillium lilacinum* | HQ842824 | - |

(Continued)

| Cult. Collec. ID | Identification | GenBank accession number | |
|---|---|---|---|
| | | **ITS** | **β-tubulin** |
| CBS 328.93 | *Purpureocillium lilacinum* | HQ842816 | - |
| CBS 128764 | *Purpureocillium lilacinum* | MH865073 | - |
| CBS 129077 | *Purpureocillium lilacinum* | MH865165 | - |
| CBS 346.51 | *Purpureocillium lilacinum* | MH856891 | - |
| CBS 119377 | *Purpureocillium lilacinum* | DTO 008-H8/038-B7* | - |
| CBS 432.87 | *Purpureocillium lilacinum* | HQ842819 | AY624228 |
| 114 B | *Purpureocillium lilacinum* | KY471671 | KY488511 |
| IHEM 18252 | *Purpureocillium lilacinum* | - | FJ515301 |
| CBS 128677[T] | *Purpureocillium lavendulum* | MH864976 | - |
| CBS 128678 | *Purpureocillium lavendulum* | MH864977 | - |
| CBS 128893 | *Metarhizium marquandii* | MH865143 | - |
| CBS 182.27[T] | *Metarhizium marquandii* | MH854923 | KJ388548 |
| CBS 130230 | *Metarhizium marquandii* | MH865781 | - |
| CBS 127148 | *Metarhizium marquandii* | MH864440 | - |
| CBS 129411 | *Metarhizium marquandii* | MH865307 | - |
| CBS 282.53 | *Metarhizium marquandii* | MH857200 | - |
| ATCC 10525 | *Metarhizium marquandii* | KP780439 | - |
| CBS 807.71 | *Metarhizium rileyi* | MH860365 | - |
| CBS 424.51 | *Metarhizium rileyi* | MH856926 | - |
| CBS 806.71 | *Metarhizium rileyi* | MH860364 | AY624250 |
| ARSEF 936 | *Metarhizium rileyi* | - | KJ398566 |
| UAMH 11176 | *Metarhizium granulomatis* | HM195306 | KJ398541 |
| UAMH 11028[T] | *Metarhizium granulomatis* | NR_132013 | KJ398540 |
| CBS 156.65 | *Isaria (=Cordyceps) farinosa* | MH858528 | - |
| CBS 111113 | *Isaria (=Cordyceps) farinosa* | AY624181 | AY624219 |
| HG 30 | *Isaria (= Cordyceps) fumosorosea* | KU523252 | - |
| ARSEF 1644 | *Isaria (= Cordyceps) fumosorosea* | EU553327 | - |
| ATCC 16312 | *Isaria (= Cordyceps) fumosorosea* | FJ269352 | - |
| CBS 244.31 | *Isaria (= Cordyceps) fumosorosea* | - | KY574472 |
| MBC 053 | *Isaria (= Cordyceps) fumosorosea* | - | KY574473 |
| CBS 105.73[T] | *Isaria (=Cordyceps) ghanensis* | AY624185 | AY624223 |
| CBS 338.51 | *Paecilomyces variotii* | FJ389930 | FJ390007 |
| CBS 102.74 | *Paecilomyces variotii* | - | EU037073 |
| CBS 110431 | *Paecilomyces variotii* | EU037054 | EU037072 |
| CBS 132734 | *Paecilomyces variotii* | JX565087 | JX565093 |
| CBS 229.73[T] | *Paecilomyces niphetodes* | NR_103580 | AY624231 |
| CBS 364.76 | *Paecilomyces niphetodes* | AY624192 | AY624230 |
| CBS 448.69 | *Paecilomyces penicillatus* | MH859348 | AY624232 |

**Notes:**
* Code from CBS database.
[T] Used for sequences of type strain cultures.
The table presents the sequences obtained from the strains isolated in this study, identified by the LSM code, and also the sequences collected from NCBI and CBS databases.

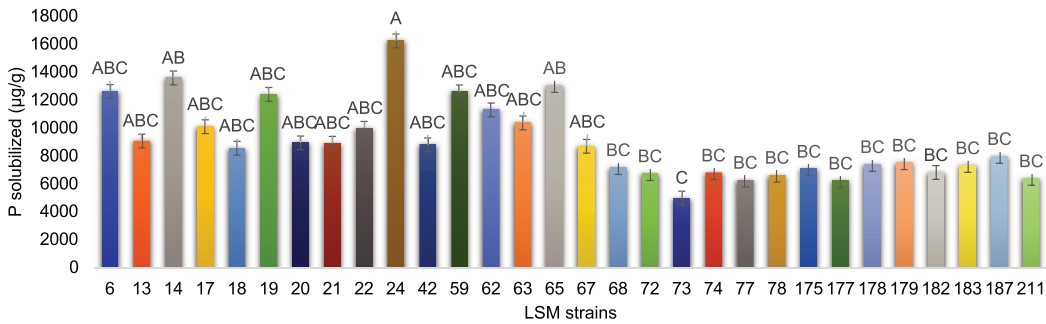

**Figure 4 In vitro P solubilization by the 30 fungal strains.** Graphic representation of fluorapatite solubilization by the 30 strains tested. Data are presented as μg of solubilized P/g of mycelium produced. The variance analysis (*F* test) was followed by Tukey's test for means comparison at 5% of probability (α = 0.05). Different letters indicate statistical significance.

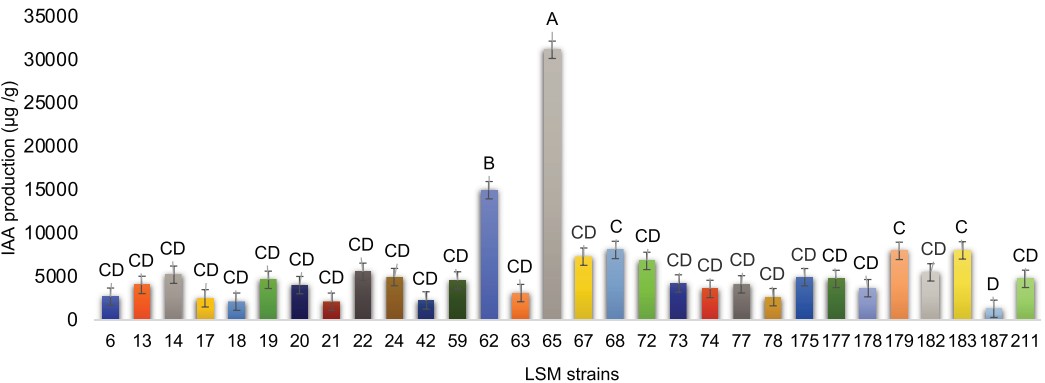

**Figure 5 In vitro IAA production by the 30 fungal strains.** Graphic representation of IAA production by the 30 strains tested. Data are presented as μg of IAA/g of mycelium produced. The variance analysis (*F* test) was followed by Tukey's test for means comparison at 5% of probability (α = 0.05). Different letters indicate statistical significance.

and benomyl in the medium. These are chemical fungicides that inhibit the growth of other highly sporulating fungi, especially *Trichoderma*, and thus favor the selective growth of *Purpureocillium*. As these chemicals are restricted by Brazilian law, they were not used, which is probably reflected in the recovery of only five strains of interest from this medium.

In the case of the DOC2 medium, *Shimazu & Sato (1996)* described it as favorable for *Beauveria bassiana* isolation. However, *Chen et al. (2010)* isolated the new species *Paecilomyces echinosporus* using DOC2 medium. It is a very poor medium, with high pH (pH = 10) and high copper concentrations. Surprisingly, this medium presented the optimal selectivity for the target fungi, and the largest number of strains of interest (23) was obtained from it. Thus, in addition to being considered for the selective isolation of *B. bassiana*, DOC2 can also be used for the selective isolation of *Purpureocillium* from soil samples.

Regarding the molecular characterization of the strains, as shown in Fig. 1, it is possible to conclude from the ITS analysis that LSM 18, 42, 73 and 211 correspond to *Purpureocillium lavendulum*, LSM 68 is a strain of *Metarhizium marquandii*, and the
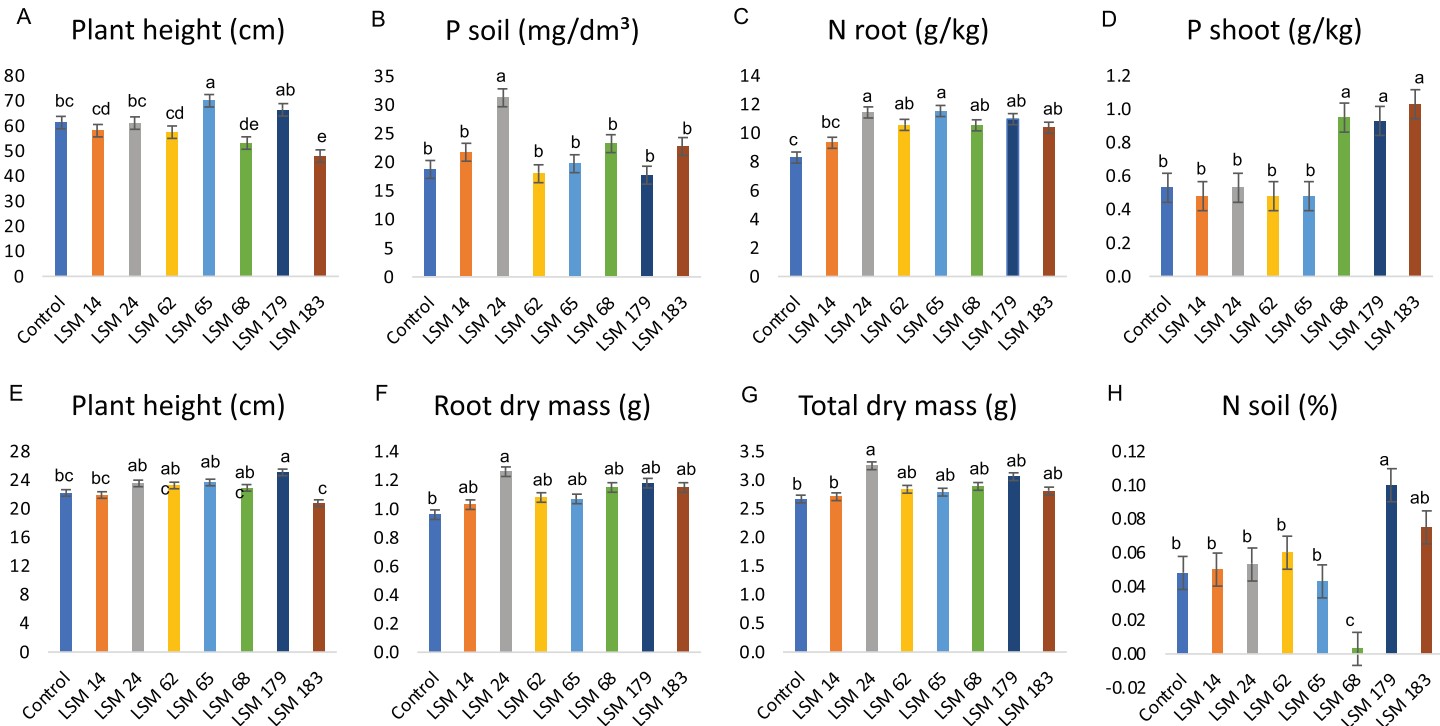

**Figure 6 Performance of the selected strains in greenhouse experiment.** Graphic illustration represents the statistical analysis of parameters evaluated. (A–C) depict results of the inoculation of *P. lilacinum* (LSM 14, 24, 62, 65, 179 and 183) and *M. marquandii* (LSM 68) strains on maize. (D) Shows the P increase in bean plants shoot and (E–H) the effect of fungal inoculation in soybean plants. The variance analysis (*F* test) was followed by Duncan's test for means comparison at 5% of probability (α = 0.05). Different letters indicate statistical significance.

remaining strains correspond to *P. lilacinum*. The same is shown in the phylogram of βTUB2 (Fig. 2), except that there are no sequences of this gene for *P. lavendulum*, which makes the identification of LSM 18, 42, 73 and 211 implicit. However, despite the absence of βTUB2 sequences for *P. lavendulum*, these four strains clustered outside of the *P. lilacinum* strains, as in the ITS analysis, indicating that it is a distinct and closely related species. The concatenated analysis of the ITS and βTUB2 sequences (Fig. 3) confirmed the pattern observed for the analysis of the two separate genes, which confirms the reliability of the identifications.

Luangsa-ard et al. (2011) suggested the combination *Purpureocillium lilacinum*, which is in the family Ophiocordycipitaceae (*Sung et al., 2007*). Other important taxonomic changes had already occurred, such as the return of *Paecilomyces javanicus* and *Paecilomyces farinosus* to the genus *Isaria* as the new combinations *Isaria javanica* and *Isaria farinosa*, respectively (*Luangsa-Ard, Hywel-Jones & Samson, 2004*). Later, *Kepler et al. (2017)* suggested a new combination for these two species and for *Paecilomyces fumosoroseus* (= *Isaria fumosorosea*) within the genus *Cordyceps*, becoming *C. javanica*, *C. farinosa* and *C. fumosorosea*, respectively.

The species *P. lavendulum* was recently described (*Perdomo et al., 2013*) after a polyphasic analysis of the strains characterized as *P. lilacinum*. This species is closely related to *P. lilacinum* and is grouped in the same family, Ophiocordycipitaceae. Unlike
*P. lilacinum*, *P. lavendulum* does not grow at 35 °C; it produces a yellowish pigment that diffuses in the culture medium, and its conidia are subglobous or lemoniform (*Perdomo et al., 2013*).

Until 2014, *M. marquandii* was recognized as *Paecilomyces marquandii*. *Kepler et al. (2014)* transferred the species to the genus *Metarhizium* because previous studies, such as *Luangsa-ard et al. (2011)*, had previously indicated the genetic grouping of this species with members of the family Clavicipitaceae, which includes the genus *Metarhizium*.

## Plant growth promotion by the fungi

To assess the potential of these strains for plant growth promotion, their P solubilization and IAA production abilities were tested in vitro. Phosphorus is a limiting nutrient in the soil and is essential for plant development because it constitutes energy molecules, nucleic acids, and coenzymes, among other compounds of great importance at the cellular level (*Patil et al., 2012*; *Gaind, 2016*). Fluorapatite was chosen for this analysis because it is a naturally occurring rock phosphate that is abundant in Brazilian soils and in different regions of the world. In addition, it has been reported that many microorganisms are generally capable of solubilizing tricalcium phosphate, the most common insoluble P source used in in vitro tests. However, these microorganisms may not be able to solubilize other phosphates, such as fluorapatite. This occurs because tricalcium phosphate is an anamorphic mineral, while fluorapatite is a crystalline mineral derived from igneous rocks and, similarly to hydroxyapatite and chlorapatite, is much more resistant to microbial action (*Fontaine et al., 2016*).

*Hernandez-Leal, Carrión & Heredia (2011)* described the ability of *P. lilacinum* to solubilize calcium phosphate, while *Cavello et al. (2015)* did not detect this ability in the *P. lilacinum* strain used in their study. To date, the ability to solubilize P from fluorapatite has not been reported in the literature for any of the species identified in this study. However, all 30 strains were able to use fluorapatite as a P source. LSM 24 of *P. lilacinum* was the best at this test and was able to solubilize more than 16 mg of phosphate per gram of mycelium (Fig. 4).

The phytohormone IAA, the best-known auxin, is responsible for several effects on the development of shoots and roots, such as tropism responses, cell division, vascular tissue differentiation and the initiation of root formation (*Jaroszuk-Ściseł, Kurek & Trytek, 2014*). In addition to these benefits, IAA is a signaling molecule in many interactions between plants and microorganisms. For example, the synthesis of IAA in the rhizosphere can modify the root architecture, increasing not only the mass but also the root area; the increased root area is then colonized by microorganisms and simultaneously increases the acquisition of nutrients from the soil by the plant (*Nieto-Jacobo et al., 2017*). *Cavello et al. (2015)* were the first to report IAA production by *P. lilacinum*, describing a maximum production of 3.24 µg/mL of IAA under the conditions tested by the authors. The *P. lilacinum* strain LSM 65, which had the highest production of IAA among the 30 strains in this study, had an average production of 53.0 µg/mL of IAA, and the *P. lilacinum* strain LSM 179 had an average production of 19.5 µg/mL of IAA (Dataset S1).

*Purpureocillium lilacinum* is a nematophagous and entomopathogenic species (*Singh, Pandey & Goswami, 2013*; *Goffré & Folgarait, 2015*) capable of the endophytic colonization of plant tissues (*Bamisile et al., 2018*). The same is reported for some species of *Metarhizium*, although no studies on this subject involving *M. marquandii* have been found in the literature.

Several studies have shown that the application of *P. lilacinum* under nematode infestation conditions, especially *Meloidogyne* spp. infestations, can improve the plant response to the presence of the pathogen and even promote plant growth. As an example, in their study, *Nesha & Siddiqui (2017)* tested a strain of *P. lilacinum* and another of *Aspergillus niger* independently and together and assessed their growth promotion potential and ability to combat pathogens in carrots. The influences of fungal inoculation on plant height, dry and fresh mass, chlorophyl content in the leaves, number of galls in the roots, nematode population, and disease rates were evaluated, and the results showed that whether isolated or together, the fungi were able to positively affect the parameters related to growth promotion and negatively affect those corresponding to the action of the pathogens.

However, only two studies have tested the effect of the inoculation of *P. lilacinum* on the promotion of plant growth in the absence of pathogens. *Lopez & Sword (2015)* evaluated the effect of endophytic colonization by *P. lilacinum* and *B. bassiana* in cotton (*Gossypium hirsutum*) and demonstrated that both fungi were able to increase the dry mass and the number of flowers of the plants. *Hernandez-Leal, Lopez-Lima & Carrión (2016)* also evaluated the effect of *P. lilacinum* inoculation in oat plants in the absence of pathogens. In this study, the authors did not obtain significant results for the growth promotion parameters evaluated, including fresh and dry mass and nutrient contents, with the tested strains of *P. lilacinum*.

*Farias et al. (2018)* tested the inoculation of *P. lilacinum* in combination with four fungi, *B. bassiana*, *M. anisopliae*, *Pochonia chlamydosporia* and *Trichoderma asperellum*, and obtained positive results in the evaluation of growth promotion parameters in soybean and maize plants; however, because they were tested together, it is difficult to determine which microorganism actually contributed to plant growth.

Seven strains were selected from the in vitro screening according to the established criteria and tested in maize, bean and soybean plants under greenhouse conditions. Therefore, this study is the first to demonstrate that *P. lilacinum* strains can significantly increase growth promotion parameters when inoculated in maize, bean and soybean plants without stimuli related to the presence of plant pathogens, such as nematodes. In this study, *P. lilacinum* strains LSM 65 and 179, selected for their IAA production, were able to increase the height of maize and soybean plants, respectively (Figs. 6A and 6E). The *P. lilacinum* strain LSM 24, selected for its P-solubilization ability, promoted an increase in root and total dry mass in soybean plants (Fig. 6F). In the case of LSM 183 for maize and LSM 179 for bean plants (Tables S1 and S2), where their inoculation resulted in a decrease in plant height and biomass for the former and in plant height for the latter, it is possible to conclude that the fungi were not able to establish a positive interaction with the plants. The strains are not phytopathogens; however, when the conditions for

interaction are provided, even if the expectation is for a positive relationship, the opposite can also occur. Specifically, in these cases, the strains and the plants probably competed for nutrients, and the use of energy in this competition was reflected in the decreases in the mentioned variables.

All the strains except LSM 14 were able to significantly increase the amount of N in maize roots, especially LSM 65 and 24. In combination with K, N is the most essential nutrient for maize development (*Coelho, 2006*). Supplying N to a crop can be achieved by the application of chemical, mineral or biological fertilizers; the latter is achieved through the use of plant growth-promoting rhizobacteria (PGPR), which are able to fix atmospheric N to make it available to plants (*Arzanesh et al., 2011*; *Abbasi et al., 2011*).

For soybean and bean, which are leguminous plants, the N supply is associated with root nodulation with symbionts (e.g., *Rhizobium*). Once fixed, N is used as a raw material in the composition of organic molecules and is then mineralized again for the release of nitrogenous inorganic compounds that can be captured by plants (*Behie & Bidochka, 2014a*).

However, biological N fertilization can also be achieved through mycorrhizal fungi (*Miransari, 2011*; *Behie & Bidochka, 2014a*), especially arbuscular mycorrhizae, which are able to supply essential nutrients such as P and N and receive carbon from photosynthesis in exchange (*Bargaz et al., 2018*; *Bitterlich et al., 2018*). *McFarland et al. (2010)* suggest that up to 50% of the required N supply can be obtained by mycorrhizal associations. Fungi obtain N from the soil efficiently through the release of N in inorganic forms from organic matter (*Miransari et al., 2009*). *Atul-Nayyar et al. (2009)* demonstrated that mycorrhizal fungi are capable of increasing the mineralization of organic matter in the soil, resulting in the release of large amounts of mineralized N that are readily available to be captured by plants.

The pioneering research of *Behie, Zelisko & Bidochka (2012)* demonstrated that the ability to provide N to plants is not exclusive to mycorrhizal fungi. The authors reported the capacity for endophytic colonization in a strain of the entomopathogenic fungus *Metarhizium robertsii* and that the fungus was able to transfer N from infected and killed *Galleria mellonella* larvae to bean (*P. vulgaris*) and grass (*Panicum virgatum*) plants. Since then, new studies such as *Behie & Bidochka (2014b)* have been carried out evaluating the ability of endophytes to transfer N from insects in seven fungal species, *M. robertsii*, *M. guizhouense*, *M. brunneum*, *M. flavoviridae*, *M. acridum*, *B. bassiana* and *Akanthomyces* (= *Lecanicillium*) *lecanii* Exclude. in four crops: two dicotyledons, *G. max* (soybean) and *P. vulgaris* (beans), and two monocotyledons, *Triticum aestivum* (wheat) and *P. virgatum* (grass). The authors highlighted that the five species of *Metarhizium* and *B. bassiana* were able to kill insect larvae, endophytically colonize the plants, and transfer N from the insects to the plants.

The increase in N levels in soil and plants (shoot or root) is not a common feature associated with the inoculation of nonmycorrhizal filamentous fungi in agricultural crops and has never been reported for *P. lilacinum*. Therefore, the results of this study strongly suggest that the tested strains, specifically LSM 65 and 24 in maize and LSM 179 in soybean, may have been able to accelerate organic matter becoming available in the soil in

addition to establishing an endophytic relationship with the plants and participating in the transportation of N from the soil.

P solubilization from soil insoluble phosphates is spread widely across several groups of fungi and occurs through the production of organic acids or enzymes, that is, phosphatases and phytases (*Gaind, 2016*). The most abundant phosphate form in the soil that can be readily absorbed by plants is orthophosphate ($PO_4^{3-}$), which can also be absorbed by fungi. The transport of phosphates from fungi to plants is well characterized for arbuscular mycorrhizal fungi (*Bargaz et al., 2018*; *Bitterlich et al., 2018*). To date, little has been reported about the mechanism of P transportation from potentially endophytic fungi, such as *P. lilacinum*, to plants (*Behie & Bidochka, 2014a*). As previously mentioned, few studies describe the ability of *P. lilacinum* to solubilize P, and even fewer studies describe the action of the fungus *in planta* in relationship to P availability.

The results obtained for LSM 24 in maize (Fig. 6B) confirmed its in vitro potential, showing that this strain is able to solubilize insoluble P from the soil and make it available to plants. However, the most impressive results were those obtained with the inoculation of LSM 68 (*M. marquandii*) and LSM 179 and LSM 183 (both *P. lilacinum*), selected for IAA production, which were able to greatly increase the P content in the shoots of bean plants (Fig. 6D), suggesting a possible endophytic interaction with positive responses to P uptake.

It is interesting to analyze the differences in the ways that the strains interact with the different plant species. As previously observed, the results for maize plants corresponded with the data obtained from in vitro analysis, in which LSM 65 and 24 presented the best results for IAA production (Fig. 5) and P solubilization (Fig. 4), respectively, in addition to the increase in N, which did not occur in the bean crop. For the bean crop, LSM 183, 179 and 68, selected for IAA production (Fig. 5), positively affected the accumulation of P in the shoot (Fig. 6D). In soybean, LSM 179 and 24 showed the best potential for plant development, corroborating their in vitro potential, because LSM 179 increased plant height; at the same time, these strains demonstrated new potential, as the N supply increased due to LSM 179, and the dry matter increased due to increased IAA production from LSM 24.

In addition to the reported potential of the *P. lilacinum* strains in this study, it is important to highlight the presence of the *M. marquandii* strain LSM 68 among those selected for *in planta* assays. Until 2014, this species was classified as *Paecilomyces marquandii*, and it is a saprophytic fungus commonly found in soil. Some studies note its potential for application in the bioremediation of heavy metal-contaminated environments, as it is often isolated from this type of environment (*Slaba & Dlugonski, 2011*; *Slaba et al., 2013*), as well as its degradation of alachlor, an herbicide in the chloroacetanilide family (*Slaba et al., 2015*; *Szewczyk et al., 2015*). Studies involving *M. marquandii* are scarce. *Ahuja, Ghosh & D'Souza (2007)* and *Ahuja & D'Souza (2009)* described the ability of a strain of *M. (= P.) marquandii* to solubilize P from tricalcium phosphate and Hirapur rock phosphate with starch as the carbon source. *Posada et al. (2013)* and *Ceci et al. (2018)* also reported the ability of strains of *M. (= P.) marquandii* to solubilize P from tricalcium and iron phosphate, respectively. Therefore, this study also

contributes unpublished information linking *M. marquandii* to plant growth promotion by testing it under greenhouse conditions and obtaining impressive results regarding P solubilization and transportation in bean plants (Fig. 6D).

The divergence in the in vitro and *in planta* responses is certainly related to several factors, starting with the tested plant species. The results indicate that each plant species interacts differently with the tested strains, and this arises from aspects such as the location of the fungus, whether it remains in the rhizosphere or interacts endophytically with the plants, whether and how it induces resistance, and whether it produces phytohormones or changes their production in the plant. Thus, in addition to field tests to confirm the potential of these fungi displayed in greenhouse conditions, other tests should be performed in the future to provide a better elucidation of the fungus-plant interaction.

## CONCLUSIONS

In addition to several valuable studies that have been published in the context of plant growth promotion combined with the use of microorganisms, this study demonstrates the importance of reporting the unprecedented potential of *P. lilacinum* and *M. marquandii* strains in plants that are agriculturally relevant worldwide and that currently consume millions of tons of agrochemicals and fertilizers in their cultivation. The results herein contribute to the expansion of knowledge that can be used for the development of more sustainable agriculture and confirm that the tested strains can be further explored with the aim of their large-scale application as bioinoculants.

### Funding

This study was funded by Fundação de Amparo à Pesquisa do Estado de São Paulo (grant number 2015/17505-3) and by the Coordenação de Aperfeiçoamento de Pessoal de Nível Superior - Brasil (CAPES), Finance Code 001. The funders had no role in study design, data collection and analysis, decision to publish, or preparation of the manuscript.

### Grant Disclosures

The following grant information was disclosed by the authors:
Fundação de Amparo à Pesquisa do Estado de São Paulo: 2015/17505-3.
Coordenação de Aperfeiçoamento de Pessoal de Nível Superior - Brasil (CAPES): 001.

### Competing Interests

The authors declare that they have no competing interests.

### Author Contributions

- Noemi Carla Baron conceived and designed the experiments, performed the experiments, analyzed the data, prepared figures and/or tables, authored or reviewed drafts of the paper, and approved the final draft.

- Andressa Souza Pollo conceived and designed the experiments, performed the experiments, analyzed the data, prepared figures and/or tables, authored or reviewed drafts of the paper, and approved the final draft.
- Everlon Cid Rigobelo conceived and designed the experiments, performed the experiments, analyzed the data, prepared figures and/or tables, authored or reviewed drafts of the paper, and approved the final draft.

## Field Study Permissions

The following information was supplied relating to field study approvals (i.e., approving body and any reference numbers):

The soil samples used to perform the study were collected at a rural property belonging to Sr. Antonio Aparecido Baron.

## Data Availability

Raw data are available in the Supplemental Files.

## Supplemental Information

Supplemental information for this article can be found online at http://dx.doi.org/10.7717/peerj.9005#supplemental-information.

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
