# Peer review of "Purpureocillium lilacinum and Metarhizium marquandii as plant growth-promoting fungi"

_PeerJ, doi:10.7717/peerj.9005_

## Round 0.1 · original submission · Major Revisions

· Academic Editor

Major Revisions

Please address the issues highlighted by our reviewers. While I do not share reviewer#2' negative opinion, I do think they have (among others) a point regarding the need for a better introduction and explanation of the motivation behind your research, since these strains are most know as entomopathogens and you are using them is a pest-free environment.

·

Basic reporting

1- Luangsa-ard et al. (2011), and Jaroszuk-Scisel et al. (2014) are not found in the “References”.
2- Line 126: Change to “potato dextrose broth (PDB)” not “dextrose potato broth”.
3- It would be useful to collect the statistical analysis of all experiments in one part under Materials and Methods section.
4- All information necessary to understand the table and figure should be included in the title or caption (especially Figure 6).

Experimental design

1- The experiment is well planned and results are interpreted in the light of the latest literature.

Validity of the findings

1- The work of the authors is well organized and the results are significant.

Reviewer 2 ·

Basic reporting

English used is not very clear but some places ambiguous.
Purpureocillium lilacinum is a common soil hyphomycete able to parasitise root knot nematode (RKN) sedentary stages by direct hyphal penetration and by using hydrolytic enzymes. The effectiveness of P. lilacinum in controlling RKNs has been reported under controlled conditions and in pot tests, but few reports are available on its effectiveness under field conditions. Purpureocillium lavendulum, which has vinaceous colonies similar to those of P. lilacinum. It is characterized by the lack of growth at 35 C, the production of a yellow diffusible pigment and by subglobose or limoniform conidia. P. lavendulum is a natural enemy of insects and plant-parasitic nematodes, and has been used as an important bio-control agent against agricultural pests. The Metarhizium genus comprises mostly entomopathogenic fungi: some are generalists, while some are specialists. The understanding of the precise mechanism of infection, its virulence and pathogenicity, as well as the enzymes and toxins produced are not well known.
Therefore work on the Purpureocillium lilacinum, P. lavendulum and Metarhizium marquandii as plant growth promoting fungi is really questionable?
Although authors have made tremendous efforts in fungal isolation, molecular and phylogenetic analysis, phosphorus solubilization assay, indoleacetic acid (IAA) production assay, mass production of selected strains and conducted greenhouse experiments but I still have doubts about the role of these fungi as plant growth promoters.
Sufficient introduction and background how this work is relevant with prior literature has not been appropriately referenced. The submission has not included all results relevant to the hypothesis.

Experimental design

Research is original but research questions are not well defined and meaningful. Research does not fill the knowledge gap. Most of researches on above mentioned fungi demonstrate that these fungi have role in biocontrol of plant diseases but this manuscript try to demonstrate the role of these fungi in plant growth promotion.
The investigation must have been conducted rigoruously both in green house and field condition to demonstrate that the role of these fungi as plant growth promoters. The research must have also been conducted in conformity with the ethical standards in the field. Experiment must have been conducted with 5 replication and repeated once.

Validity of the findings

Unless field experiments are not performed under field condition the conclusions drawn are not acceptable. Drawn conclusion are limited to those supported by the results.

Additional comments

English used is not very clear but some places ambiguous.
Purpureocillium lilacinum is a common soil hyphomycete able to parasitise root knot nematode (RKN) sedentary stages by direct hyphal penetration and by using hydrolytic enzymes. The effectiveness of P. lilacinum in controlling RKNs has been reported under controlled conditions and in pot tests, but few reports are available on its effectiveness under field conditions. Purpureocillium lavendulum, which has vinaceous colonies similar to those of P. lilacinum. It is characterized by the lack of growth at 35 C, the production of a yellow diffusible pigment and by subglobose or limoniform conidia. P. lavendulum is a natural enemy of insects and plant-parasitic nematodes, and has been used as an important bio-control agent against agricultural pests. The Metarhizium genus comprises mostly entomopathogenic fungi: some are generalists, while some are specialists. The understanding of the precise mechanism of infection, its virulence and pathogenicity, as well as the enzymes and toxins produced are not well known.
Therefore work on the Purpureocillium lilacinum, P. lavendulum and Metarhizium marquandii as plant growth promoting fungi is really questionable?
Although authors have made tremendous efforts in fungal isolation, molecular and phylogenetic analysis, phosphorus solubilization assay, indoleacetic acid (IAA) production assay, mass production of selected strains and conducted greenhouse experiments but I still have doubts about the role of these fungi as plant growth promoters.
Sufficient introduction and background how this work is relevant with prior literature has not been appropriately referenced. The submission has not included all results relevant to the hypothesis.
Research is original but research questions are not well defined and meaningful. Research does not fill the knowledge gap. Most of researches on above mentioned fungi demonstrate that these fungi have role in biocontrol of plant diseases but this manuscript try to demonstrate the role of these fungi in plant growth promotion.
The investigation must have been conducted rigoruously both in green house and field condition to demonstrate that the role of these fungi as plant growth promoters. The research must have also been conducted in conformity with the ethical standards in the field. Experiment must have been conducted with 5 replication and repeated once.

Unless field experiments are not performed under field condition the conclusions drawn are not acceptable. Drawn conclusion are limited to those supported by the results.

·

Basic reporting

The English language should be thoroughly revised by a native speaker. There are numerous instances of misuse of prepositions, verbal tenses and conjunctions. At times the authors employ informal language, i.e., “specially” instead of “especially” (line 23) or phrasal verbs, “it did not stand out” (line 281, which also exemplifies my point about tenses).
The background provided is sufficient, although it is a little hard to follow when trying to introduce the changes in the genus from Paecilomyces to Purpureocillium; I suggest reordering this information. In addition, I recommend including the following references: three previous reports of the genus Purpureocillium/Paecilymoces as natural endophites (Bills & Polishook 1991 Microfungi from Carpinus caroliniana, Can J Bot 69, 1477-1482; Cao et al. 2002, Endophytic fungi from Musa acuminata leaves and roots in South China, W J Microbiol Biotech 18, 169-171; Tian et al. 2004, Study on the communities of endophytic fungi and endophytic actinomycetes from rice and their antipathogenic activities in vitro, W J Microbiol Biotech 20, 303-309).
The in-text citation style should be revised to follow that recommended by the journal and/or to maintain the same format throughout the text.
The structure of the article conforms to the format of standard sections, and the headlines improve the clarity of the manuscript.
The figures are relevant to the content of the article, but the labels of Figures 4, 5 and 6 are incomplete. Following the Instructions for Author, the journal requires a title for each figure, and all figures seem to have only a legend, not a title with a legend (when the latter is optional).
Available raw data is correct.

Experimental design

The manuscript clearly defined the potential use of microorganisms as bioinoculants that promote plant growth, and evaluated different strains of three species of which there is little to no information regarding this subject in three different crops (maize, bean and soybean). The solubilization of phosphorus and the production of a component related to plant promotion, indoleacetic acid, were also evaluated in screening 30 strains, among which the best ones were selected. After that, greenhouse tests were done and growth promotion parameters including plant height, dry mass and contents of P and nitrogen (N) in the plants and in the rhizospheric soil were assessed. Methods are sufficiently described, although some further details necessary for reproducibility would be: 1. reporting the manufacturer of the Wizard SV Gel and PCR Clean-up System (line 130), 2. indicating in which culture medium the viability test was performed (line 220).

Validity of the findings

Discussion of the results is well stated, connected with the main question addressed by the research and limited to which is supported by the results, although I think it is a bit rash to establish an endophytic relationship only with the data obtained and without an experimental intervention that supports it (lines 469-470).
Although the medium for the isolation of fungi was not an original question, I consider it appropriate and relevant to show and discuss these results.
It is important to highlight the demonstration that some evaluated strains were able to promote an increase in plant growth. However, it is not entirely clear to me that the previous screening has a relevant influence on the results, especially when the strains selected showed different results in different plants, and the best strains were selected from a different ability (P-solubilization or IAA production). I believe it would be more appropriate to include at least one strain that had the opposite result to validate the screening. With these results, there is still the chance that some of the other 24 species not evaluated could be better for plant promotion.
On the other hand, it is necessary to include at least some comments on those strains that seemed to produce a detrimental effect on plants, for example the strain LSM 179 on beans and LSM 183 on maize. The former was the strain that produced better results in soybean plants and thus it needs to be mentioned. Differences in the way that the strains interact with the different plant species are mentioned but only for the positive results.
Other comments: 1) it is not true that P level in the roots of the control treatment was higher than in the others; table S2 shows no significant differences among treatments and the absolute value obtained in the control is not the highest (lines 291-292); 2) the N content in soil for soybean plants was significantly different among treatments and also had a high coefficient of variation; thus, this does not seem to be the reason of the absence of statistical differences in the other variables mentioned (lines 312-314).

Additional comments

Section “In vitro screening tests and selection of strains for in planta tests”: there are similar sentences throughout this section. I suggest to include all the information related to P solubilization and IAA production in only one paragraph to avoid repetitions.
Lines 272 and 487: It should read Fig. otherwise FIG.
Line 274: Change supplmental with supplemental
Line 305: the font color of this sentence seems to be darker.
Lines 430 and 432: It should say Fig.6 instead of Fig.1 in all cases.
Line 618: It should say “Goffré” instead of “Goffre”
Table 1: It should say “Table 1” instead of “Tabel 1”

---

## Round 0.2 · accepted · Accept

· Academic Editor

Accept

I am satisfied with your responses to the issues raised by our reviewers.

·

Basic reporting

The manuscript was greatly improved.

Experimental design

The present work was organized logically, and the results obtained were reliable and persuasive.

Validity of the findings

A great deal of work was conducted here, and the results were very accurate and significant.

Additional comments

The authors have satisfactorily responded to all the raised concerns and comments by the reviewers. The manuscript is now suitable for acceptance from my point of view.

·

Basic reporting

The English language of the article is now clear and technically correct.
The background provided is sufficient.
The in-text citation style follow that recommended by the journal.
The structure of the article conforms to the format of standard sections, and the headlines improve the clarity of the manuscript.
The figures are relevant to the content of the article. However, P solubilization graphic should be the Figure 4, then “In vitro IAA production...” should be the Figure 5, and the last should be the Figure 6. Probably, there was an error in the submission process.
Available raw data is correct.

Experimental design

Methods are now sufficiently described.

Validity of the findings

Discussion of the results is well stated, connected with the main question addressed by the research and limited to which is supported by the results.
The previously identified issues were correctly argued and clarified, with an exception: the authors mentioned a change in lines 476-478 (“probably by” instead of “in addition to”) that was not observed in the final version.

Additional comments

Table 1 (legend at the end): It should say “Table 1” instead of “Tabel 1”